# Tungsten Diselenide Nanoparticles Produced via Femtosecond Ablation for SERS and Theranostics Applications

**DOI:** 10.3390/nano15010004

**Published:** 2024-12-24

**Authors:** Andrei Ushkov, Dmitriy Dyubo, Nadezhda Belozerova, Ivan Kazantsev, Dmitry Yakubovsky, Alexander Syuy, Gleb V. Tikhonowski, Daniil Tselikov, Ilya Martynov, Georgy Ermolaev, Dmitriy Grudinin, Alexander Melentev, Anton A. Popov, Alexander Chernov, Alexey D. Bolshakov, Andrey A. Vyshnevyy, Aleksey Arsenin, Andrei V. Kabashin, Gleb I. Tselikov, Valentyn Volkov

**Affiliations:** 1Moscow Center for Advanced Studies, Kulakova Str. 20, Moscow 123592, Russia; ushkov.andrei.a@gmail.com (A.U.); dmitriydyubo@gmail.com (D.D.); nmbelozerova@jinr.ru (N.B.); dmitriiy911@gmail.com (D.Y.); alsyuy@xpanceo.com (A.S.); ditselikov@mephi.ru (D.T.); martinov@mitht.org (I.M.); bolshakov@live.com (A.D.B.); vyshnevyy@xpanceo.com (A.A.V.); alekseyarsenin@ysu.am (A.A.); 2Frank Laboratory of Neutron Physics, Joint Institute for Nuclear Research, Dubna 141980, Russia; 3Emerging Technologies Research Center, XPANCEO, Internet City, Emmay Tower, Dubai, United Arab Emirates; kazantsev.is@xpanceo.com (I.K.); tikhonowski@xpanceo.com (G.V.T.); ermolaev-georgy@xpanceo.com (G.E.); grudinin@xpanceo.com (D.G.); celikov@xpanceo.com (G.I.T.); 4MEPhI, Institute of Engineering Physics for Biomedicine (PhysBio), Moscow 115409, Russia; aapopov1@mephi.ru; 5Faculty of Physics, St. Petersburg State University, Universitetskaya Emb. 7–9, St. Petersburg 199034, Russia; 6Center for Nanotechnologies, Alferov University, Khlopina 8/3, St. Petersburg 194021, Russia; 7Laboratory of Advanced Functional Materials, Yerevan State University, Yerevan 0025, Armenia; 8National Center for Scientific Research, LP3, Aix-Marseille University, CNRS, 13288 Marseille, France; andrei.kabashin@univ-amu.fr

**Keywords:** tungsten diselenide, laser ablation, nanoparicles, optical scattering, photoheating, SERS

## Abstract

Due to their high refractive index, record optical anisotropy and a set of excitonic transitions in visible range at a room temperature, transition metal dichalcogenides have gained much attention. Here, we adapted a femtosecond laser ablation for the synthesis of WSe_2_ nanoparticles (NPs) with diameters from 5 to 150 nm, which conserve the crystalline structure of the original bulk crystal. This method was chosen due to its inherently substrate-additive-free nature and a high output level. The obtained nanoparticles absorb light stronger than the bulk crystal thanks to the local field enhancement, and they have a much higher photothermal conversion than conventional Si nanospheres. The highly mobile colloidal state of produced NPs makes them flexible for further application-dependent manipulations, which we demonstrated by creating substrates for SERS sensors.

## 1. Introduction

Tungsten diselenide (WSe_2_) belongs to a family of van der Waals materials that have attracted exceptional attention since the isolation of graphene in 2004 [1] due to their extraordinary electrical, optical and magnetic properties. Being a transition metal dichalcogenide (TMDC), WSe_2_ has a layered crystalline structure, where neighboring hexagonal, closely packed monolayers of tungsten and selenium are stacked by weak van der Waals (vdW) forces (see Figure 1a). The unique material structure of TMDCs leads to their outstanding mechanical properties (exfoliation down to atomically thin flakes [2,3]), electronic (electronic structure engineering [4]) and optical (high refractive index and giant anisotropy [5,6]) characteristics, which drive extensive TMDC research for medical [7,8,9], electronic [10,11], optical [12,13,14,15] and space [16] scientific communities.

Among a set of common transition metals of TMDCs, the tungsten atom is the heaviest, which sufficiently impacts the WSe_2_ properties. For example, the WSe_2_ monolayer valence band splitting is three times larger than that for the MoS_2_ monolayer, which makes it promising for valleytronic and spintronic applications [17,18]. Unlike most TMDC materials [19,20,21,22], which exhibit n-type conductivity without doping, WSe_2_ is hole-conducting and is among the most frequently used p-type 2D materials [23,24]. In contrast to WSe_2_’s single- and few-layered state, its bulk form is much less elaborated in terms of applications, despite the abundance of shapes already reported: nanosheets assembled into nanoflowers [25], nanotubes [26], vertically aligned nanosheets [27] and quantum dots [28]. Notably, bulk WSe_2_, however, has a set of advantageous characteristics for optics, inherited from the TMDC material family: a high refractive index n>4 in the visible and infrared spectral ranges, a prominent (20–30%) optical anisotropy and a set of excitonic transitions in the visible range at room temperature. Importantly, WSe_2_ has a resonant excitonic absorption peak at ≈750 nm, which is extremely interesting for photoheating and theranostic applications in the NIR-I transparency window of biological tissues. The key issue of exploiting the high refractive index of WSe_2_ in photonic nanostructures is the efficiency and output quality of fabrication methods.

Femtosecond laser material processing is a high-throughput manufacturing tool employed in science and industry and offers the advantage of a flexible, precise treatment of a wide range of materials [29,30,31].

For van der Waals materials, naturally occurring in a layered plane geometry, their shape transformation into spherical nanoparticles is a challenging task [32]. It was already shown that the femtosecond pulsed laser ablation in liquid (PLAL) approach allows the synthesis of spherical vdW NPs, which inherit the crystalline and optical properties of the bulk target, demonstrate an onion-like or polycrystalline structure and have a broad size distribution from 5 to 250 nm [33,34,35]. For the first time, it was demonstrated for MoS_2_ and WS_2_ materials [33,34], then for MoSe_2_ as well [35]. Interestingly, in the case of MoS_2_ NPs, the laser fragmentation can be employed to further tune their atomic composition and yield oxidized MoO_3−x_ NPs [34]. In the case of the MoSe_2_ target, the femtosecond ablation revealed the 2H/1T phase transition in NPs [35].

In our work, we study the possibilities and features of the synthesis of spherical WSe_2_ nanoparticles by the femtosecond laser ablation method. Previously, we have successfully used PLAL for preparing NPs of Si [36], TiN [37], MoS_2_ [33,34] and other materials, including core-satellite nanocomposites [38,39,40]. Here, we demonstrate that PLAL yields crystalline WSe_2_ NPs with a nearly spherical shape and diameters ranging from ∼10 nm to ∼150 nm. Importantly, prepared NPs conserve the high refractive index and excitonic response of an initial bulk WSe_2_ target. The latter is illustrated via the photoheating experiments with a tunable Ti–sapphire laser in a wavelength range from 700 nm to 870 nm, which passes through the WSe_2_ A-exciton position (∼750 nm). We believe that the proposed high-yield and substrate-free technique will unlock new optical and theranostic applications of WSe_2_ NPs.

## 2. Materials and Methods

### 2.1. Sample Preparation

Femtosecond pulsed laser ablation in liquids (PLAL) was used to synthesize WSe_2_ nanoparticles (NPs). A high-purity, synthetically grown bulk WSe_2_ crystal (2D Semiconductors Inc., Scottsdale, AZ, USA) with dimensions of 1 × 4 × 4 mm served as the target material. The crystal was placed inside a 10 mL glass vessel filled with 2 mL of DI water. The height of the water column above the crystal surface was 10 mm and was kept constant in all experiments. A femtosecond Yb:KGW laser system (TETA-10, Avesta, Moscow, Russia) operating at a wavelength of 1033 nm was employed for the ablation process. The laser delivered pulses with a duration of 270 fs, a pulse energy of 100 μJ, and a repetition rate of 1 kHz. The laser beam was focused by a lens with a working distance of 100 mm. The lens-to-crystal distance was carefully adjusted to between 95–105 mm to maximize the efficiency of nanoparticle synthesis. Beam scanning in the XY plane across the target surface was achieved using two mutually perpendicular linear translation motorized stages, which perform a cyclic sweep of a square pattern at a speed of 5 mm/s. A standard ablation duration was set at 10 min. After ablation, the prepared colloidal solutions of WSe_2_ NPs were collected and analyzed without further purification or chemical treatment, except differential centrifugation.

### 2.2. Sample Characterization

The crystalline structure of single NPs was characterized by the high-resolution TEM system (JEM 2010; JEOL, Tokyo, Japan) at 200 kV. Drops of 2 µL of sample colloids were drop-casted on carbon-coated TEM copper grids and dried at ambient conditions.

A scanning SEM system (MAIA 3; Tescan, Brno, Czech Republic) with an integrated EDX detector (X-act; Oxford Instruments, Abingdon, UK) was used to examine the shape of large NPs at an inclined view and EDX characterization of the bulk crystal ablation target. The samples were prepared by dropping 2 µL of colloids on a clean Si substrate and drying at ambient conditions.

For temperature-dependent Raman characterization, a 20 µL drop of a sample colloid was drop-casted on a cover glass substrate instead of a metallic substrate to avoid the heat transfer to the substrate and dried at ambient conditions for 1 h.

For SERS measurements, 20 µL drops of sample colloids were drop-casted on aluminum surface and spin-coated to obtain thin films. As-prepared SERS substrates were covered with 2 µL of dye of various concentrations $10^{-4}$–$10^{-8}$ M and dried under ambient conditions for 1 h. Raman spectroscopy was performed using Horiba LabRAM HR Evolution setup with excitation wavelengths of 532 nm and 633 nm, a diffraction grating of 600 lines/mm and a microscope objective of 100×, NA 0.9. A good reproducibility of spectra was observed during the measurements.

## 3. Results and Discussion

### 3.1. Laser Ablation Synthesis of WSe_2_ NPs

Bulk tungsten diselenide has a layered structure typical for TMDCs materials. Inside every layer, tungsten atoms have a trigonal prismatic coordination with regard to selenium; therefore, the hexagonal lattice visible from the top is composed of selenium, tungsten and, again, selenium sandwich planes (see Figure 1a). The distance between neighboring sandwich layers is ≈3.14 Å [41].

The PLAL setup is schematically shown in Figure 1d and in Appendix A. The bulk crystal is placed at the bottom of the glass vessel, filled with DI water. The free surface of the liquid is 1 cm above the top surface of the crystal. Femtosecond laser pulses (1030 nm, 280 fs, 100 μJ) are focused on the bulk crystal surface via the lens (focal distance 100 mm) and generate cavity bubbles with ionized plasma. The plasma cooling and cavity bubbles deflation leads to the plasma condensation into crystalline NPs of different size. The prepared colloid demonstrates a good stability (during at least one month after the synthesis) due to the significant charge of WSe_2_ NPs with an electrostatic potential reaching −30…−40 mV (see Appendix A).

Bulk WSe_2_ crystal for PLAL was purchased from 2D Semiconductors Inc. The energy-dispersive X-ray spectroscopy (EDX) of the crystal (Figure 1b) confirms the stoichiometry quality, W (32.52 at.%), Se (67.48 at.%), whereas the selected-area electron diffraction (SAED) of microscopic WSe_2_ flake demonstrates a typical hexagonal pattern of monocrystalline TMDCs (Figure 1c).

### 3.2. Morphology of WSe_2_ Nanoparticles

The structural characteristics of WSe_2_ nanoparticles synthesized by pulsed laser diffraction (PLAL) in liquids are presented in Figure 2. Transmission electron microscopy (TEM) image (Figure 2a) shows the distribution of the nanoparticles. Most of them have a near-spherical morphology. To further confirm that, we performed a tilted scanning electron microscopy (SEM) (Figure 2b) at an inclination angle of 45 degrees and found that NPs retain a round shape in the image. The EDX spectrum (Figure 2c) confirms that NPs conserve the atomic composition of the original crystal: W (12.95 at.%) and Se (29.88 at.%). Some additional peaks correspond to the TEM grid copper Cu (57.17 at.%). The SAED pattern (Figure 2d) reveals distinct diffraction rings, which correspond to the interplanar spacings of crystalline WSe_2_. The distances corresponding to the planes (0 0 4), (1 0 0), (0 0 6), (1 0 4), (1 0 6) and (1 1 0) are in a good agreement with theoretically calculated values (see Appendix A). High-resolution TEM (HRTEM) of a single WSe_2_ NP (Figure 2e) further confirms its polycrystalline structure; there are visible zones with different orientations of crystallites. The differential centrifugation of the synthesized colloid was performed in order to separate various average NPs sizes (see the next Section). The fact that the Raman spectra of the as-prepared colloids (Figure 2f) demonstrate a characteristic WSe_2_ peak at 250 cm^−1^ and, generally, repeat spectral features of the bulk crystal indicates the quality of WSe_2_ NPs in a broad range of sizes.

### 3.3. Optical Response of WSe_2_ Nanoparticles

The PLAL approach inherently produces a polydisperse colloid. Remarkably, the colloid remained stable for over a month with negligible aggregation/precipitation, due to a significant negative zeta potential ≈35 mV (see Appendix A). Given the high refractive index of WSe_2_ in the visible spectrum, pronounced changes in optical properties may be achieved due to variations in nanoparticle size. To obtain solutions with different average diameters of NPs, the initial colloid underwent differential centrifugation. Particle size distributions obtained from the manual analysis of TEM images and dynamic light scattering (DLS) measurements confirmed log-normal size distributions, with the average particle size shifting to smaller values with increasing revolutions per minute (rpm). Taking into account that the size distribution of initial colloid is log-normal as well, it leads to different mass concentrations of samples and, consequently, to different color saturations (transparencies) of the solutions at different centrifugation speeds (Figure 3e). The most transparent solutions were obtained for lowest and highest centrifugation speeds (see Figure 3e). Moreover, the difference in solutions’ hues indicates the difference in their extinction properties (see Figure 3b). Firstly, solutions with smaller NPs possess a more rapid extinction fall at a shorter wavelength ($1/\lambda^{4}$), typical for the Rayleigh scattering regime. Secondly, the excitonic peak, located at ≈750 nm for colloids with NPs of smaller average size, is red-shifted for larger NPs (see the inset in Figure 3b). The presence of the excitonic peak, which originates from the bulk crystal (see Figure 3d), indicates NPs’ crystallinity. The polycrystalline nature of NPs is taken into account in extinction calculations by using a homogenized dielectric function, $\varepsilon_{av}=2\varepsilon_{o}/3+\varepsilon_{e}/3$, where $\varepsilon_{o}$ and $\varepsilon_{e}$ are in-plane and out-of-plane bulk dielectric permittivities, correspondingly. The peak shifting is caused by the fact that for WSe_2_ NPs large enough (i.e., 130 nm in diameter), the total extinction peak consists of two components (see Figure 3c): a material excitonic feature (visible as a peak in electrical total extinction contribution) and a magnetic dipole resonance (visible in magnetic dipole channel). For WSe_2_ NPs large enough, the magnetic dipole resonance, as well as other magnetic and electric multipoles, shifts towards longer wavelengths with increasing the particle size, which is an important feature of the Mie scattering regime. Thus, the synthesized particles might have applications not only as Rayleigh scatterers, but as Mie-resonators as well after a more methodical separation of large WSe_2_ NPs.

### 3.4. Photothermal Applications of WSe_2_ Nanoparticles

WSe_2_, as an efficient absorber in visible and near-IR bands, has gained much attention as a perspective thin film solar cell absorber [42,43,44]. In this regard, we examine the photothermal properties of WSe_2_ NPs in visible range at a wavelength of 532 nm, which is readily accessible in our Raman setup and is close to the peak wavelength of solar irradiance at Earth’s surface [45].

Figure 4a demonstrates a temperature-dependent shift in range from 25 ℃ up to 300 ℃ of Raman $E_{2g}^{1}$ peak of WSe_2_ NPs, drop-casted on a cover glass substrate instead of a metallic substrate to avoid the heat transfer to the substrate. The sample temperature was monitored via the LINKAM temperature control stage; Raman excitation was set 20 μW at 532 nm. We assume a linear dependence of the Raman peak position ω(T) on temperature *T* due to the relatively small temperature range examined in the study:(1)$$\omega \left(T\right)=\omega_{0}+\chi_{1}T,$$
where $\omega_{0}$ is $E_{2g}^{1}$ peak position at 25 ℃ and $\chi_{1}\approx -0.013$ is the first-order temperature coefficient, obtained from the linear fitting of experimental data from Figure 4a. The obtained $\chi_{1}$ is within a 10% deviation tolerance with values reported in recent works on multilayered WSe_2_ flakes [46,47] (see Appendix A). Then, we use Equation (Equation 1) to retrieve the local sample temperature from temperature-dependent Raman spectra. The excitation laser 532 nm with different powers was focused into a diffraction-limited spot via 100× microscope objective, thus revealing the laser-induced heating efficiency (see Figure 4b). Interestingly, at a quite low irradiation power of 0.7 mW, WSe_2_ NPs are heated by almost 350 ℃ and, therefore, have more than four times higher photothermal efficiency in comparison with WSe_2_ bulk crystal, which originates from an enhanced nanoresonators absorption.

Photothermal antibacterial and cancer therapy nowadays attracts growing attention as a minimally invasive nanomedicine approach. Due to the bacteria’s inability to adapt and elaborate anti-thermal mechanisms, efficient nanosensitizers are highly required for successive method implementation. Thermal agents should possess a sensitivity to the NIR-I optical window, as well as have a symmetrical (spherical) shape and small (less than 100 nm) size to facilitate their uptake by biological cells. We examine the phototermal response of synthesized crystalline WSe_2_ NPs in the NIR-I optical window (700–980 nm). A collimated beam of 830 nm laser source passes through a cuvette with 1 mL of heated colloid. The solution temperature as a function of time was recorded by a thermal imaging camera. Consequent heating–cooling cycles of the same WSe_2_ colloid confirmed a good reproducibility of a photothermal conversion coefficient [48] η≈0.44 (see Figure 4c). The photothermal response of WSe_2_ NPs was compared with Si NPs with the same extinction of water solutions at 830 nm (see Figure 4d). Due to the presence of excitonic transitions in a visible range, which are red-shifted as compared with those in Si, WSe_2_ NPs possess a higher absorption component in the total extinction cross-section in near-IR (see Appendix A), which results in a 4-fold enhancement of the photothermal efficiency (see Figure 4e).

Photoheating experiments analogous to those presented in Figure 4e were performed with a tunable titanium-sapphire laser source for wavelengths in the range from 700 nm to 870 nm, which includes the WSe_2_ A-exciton (see Figure 4f.) The experimentally observed photothermal conversion efficiency η is 0.48 for the shortest utilized wavelength of 700 nm, reaches the maximum value of 0.55 close to the excitonic resonance at 750 nm and drops down at longer wavelengths. Colloidal extinction was measured using transmission through a cuvette with a deionized water as a baseline; the colloidal absorption curve was obtained by multiplying the values of extinction by η (see Appendix A). This approach allows retrieving colloidal optical absorption properties from photoheating experiments.

### 3.5. WSe_2_ NPs for SERS

Figure 5a,b presents Raman spectra of the rhodamine 6G (R6G) and crystal violet (CV), respectively, in the concentration range of 10^−4^–10^−8^ M, which were enhanced by means of WSe_2_ substrates.

The Raman spectra of R6G molecules acquired from the WSe_2_ substrate exhibit strong lines at 614, 776, 1130, 1185, 1315, 1364, 1510, 1574 and 1649 cm^−1^, respectively, with excitation at 532 nm. All peaks obtained are in good agreement with earlier Raman R6G reports [49,50].

A pronounced line on 614 cm^−1^ corresponds to the C–C–C ring in-plane bending vibration. The C–H out-of-plane bending vibration for R6G is observed at 776 cm^−1^, while the C–H in-plane bending vibration is observed at 1130 cm^−1^. The C–O–C stretching frequency appears at 1187 cm^−1^. Peaks at 1315, 1364, 1510, 1574 and 1649 cm^−1^ correspond to the aromatic C–C stretch of the R6G molecule [51].

The SERS spectrum of CV contains peaks at 1619 and 1588 cm^−1^, which correspond to the C–C stretching vibration of the phenyl ring. The peak at 1372 cm^−1^ is the C–C center stretching vibration, and those at 1175 and 804 cm^−1^ are C–H bending vibrations. The radical-ring skeletal vibration and C–N bending vibration occur at 915 and 423 cm^−1^, respectively [52,53].

As expected, the SERS intensity was gradually weakened with decreasing R6G and CV concentration. When the concentration goes down to $1\times 10^{-7}$ M, the R6G and CV peaks can still be detected in range of 400–1700 cm^−1^ with feeble intensity (see Figure 5), demonstrating that the detection limit of the WSe_2_ substrate has been reached. Current works [54,55] approve the high efficiency of SERS sensors based on WSe_2_ structures, with LOD values in the range of $10^{-5}$–$10^{-8}$ M.

We perform the analogous SERS experiments on substrates with crystalline MoS_2_ NPs, synthesized by the same PLAL approach. Interestingly, nanoparticles from both TMDCs demonstrate a pronounced SERS effect (see Figure 5). The LOD values for MoS_2_ are also moderate ($10^{-7}$ M), which might be caused by the SERS chemical mechanism (CM) [56]. In Figure 5d, MoS_2_ peaks (marked by asterisks) at low dye concentrations of $10^{-6}$–$10^{-8}$ M were observed in the range of 370–470 cm^−1^ of CV spectra. The MoS_2_ modes were identified as $E_{2g}^{1}$ at 385 cm^−1^, A_1g_ at 409 cm^−1^ and 2LA(M) at 460 cm^−1^. The appearance of these spectral lines is caused by a CV signal weakness at low dye concentrations.

The obtained structures have scope for improvement. For example, considering particles with a larger specific surface area and more intense absorption at excitation wavelengths may be attractive for SERS. Moreover, the Raman signal can be enhanced [54,57,58,59] by means of creating hybrid structures (TMDC–TMDC) or core–shell/ core–satellite structures (TMDCs-Au NPs) with a lower detection threshold.

In the broader context, PLAL-synthesized colloids from different TMDCs allow to find appropriate optical properties (i.e., refractive index, excitonic absorption), which, considering a high stability of NP solutions [33], may be desirable in applications.

## 4. Conclusions

In conclusion, we adapted the femtosecond laser ablation method for the production of WSe_2_ NPs from bulk crystal. Importantly, the synthesized NPs retain the crystalline (polycrystalline) structure of the target and inherit the high refractive index and excitonic absorption. The high photothermal response is four times stronger than that of conventional Si scatterers, which is attractive for medical purposes like cancer therapy [60], whereas the high refractive index and variety of accessible sizes unlock WSe_2_ NPs for Mie-tronics [61]. SERS sensing with WSe_2_ nanoparticles substrate was also demonstrated. Generally, highly symmetrical, nano-sized, crystalline and highly refractive NPs are promising for various applications in postsilicon nanotechnologies.

## Figures and Tables

**Figure 1 nanomaterials-15-00004-f001:**
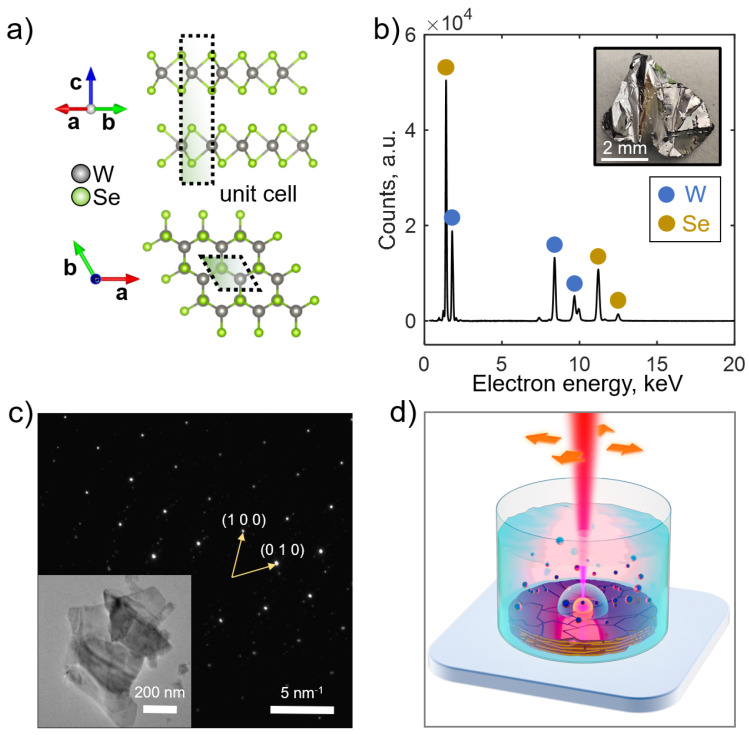
(**a**) Crystal structure of bulk WSe_2_; (**b**) EDX characterization of bulk WSe_2_ crystal; (**c**) SAED characterization of microscopic WSe_2_ flake. Yellow arrows with indices denote reciprocal lattice vectors. (**d**) Schematic view of PLAL.

**Figure 2 nanomaterials-15-00004-f002:**
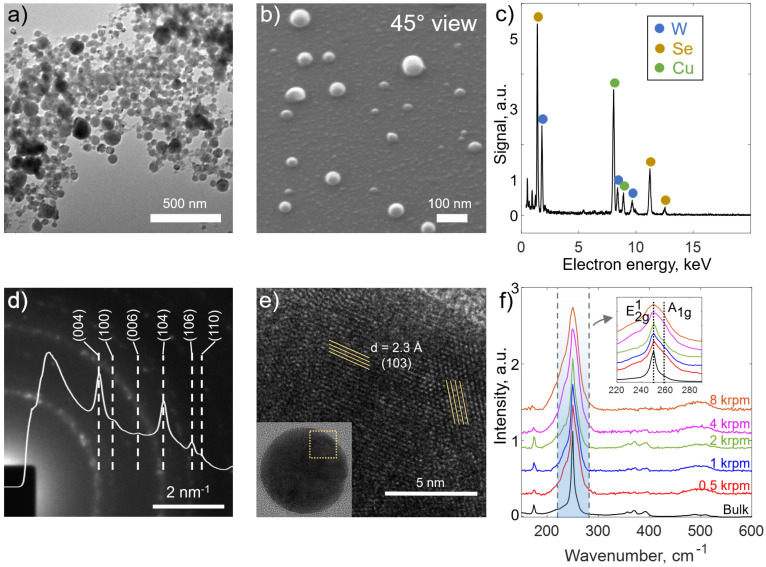
(**a**) Typical TEM image of the synthesized WSe_2_ NPs; (**b**) SEM photographs at inclined view revealing the spherical shape of NPs; (**c**) EDX spectrum from the synthesized NPs showing the elemental composition of WSe_2_ NPs. Copper signal is from the TEM grid. (**d**) SAED on synthesized WSe_2_ NPs with the most visible d_hkl_ lines; (**e**) TEM image of a single nanoparticle showing its polycrystalline structure; (**f**) Raman spectra of bulk WSe_2_ crystal and synthesized NPs, separated by centrifugation at different rotation speeds. Excitation wavelength was 532 nm.

**Figure 3 nanomaterials-15-00004-f003:**
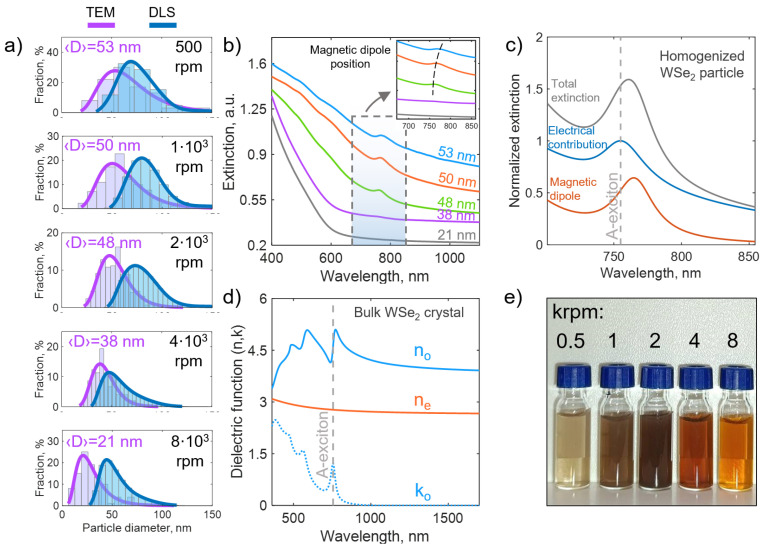
Differential centrifugation of WSe_2_ NPs. (**a**) Size distributions and average sizes of nanoparticles, obtained at different rotational speeds and measured by counting on TEM image (violet curve) and by dynamic light scattering spectroscopy (blue curve). (**b**) Measured extinction spectra for WSe_2_ colloids with various NP average diameter 〈D〉. (**c**) Calculated extinction spectra (total and contributions from electric and magnetic dipole channels) for a spherical WSe_2_ NP with a homogenized isotropic dielectric function, obtained from bulk WSe_2_ crystal data in (**d**) as $\varepsilon_{av}=2\varepsilon_{o}/3+\varepsilon_{e}/3$. (**d**) Optical constants of bulk WSe_2_; (**e**) Image of bottles with centrifugated WSe_2_ colloidal solutions in DI water.

**Figure 4 nanomaterials-15-00004-f004:**
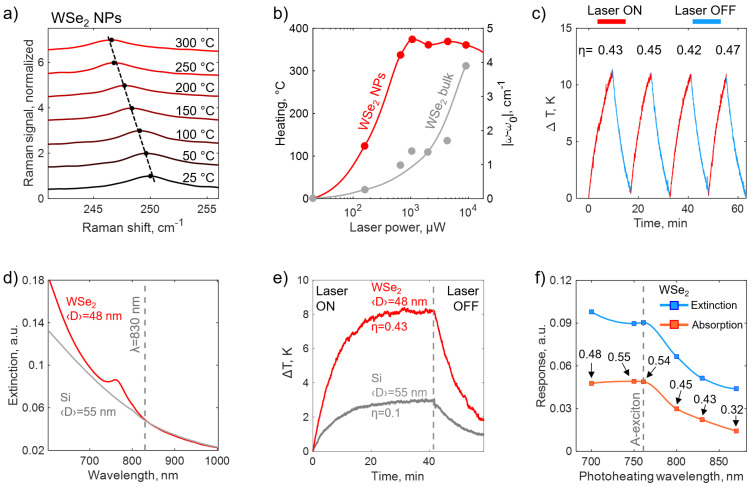
Photoheating response of WSe_2_ NPs. (**a**) Temperature-dependent Raman spectra at 532 nm excitation of WSe_2_ NPs. (**b**) Laser-induced heating (532 nm irradiation) and $E_{2g}^{1}$ peak position shift of WSe_2_ NPs and bulk crystal. (**c**) Dynamics of laser-induced heating of colloids by laser diode 830 nm, 1 W. (**d**) Measured extinction spectra of WSe_2_ and Si water colloids, normalized at photoheating wavelength 830 nm. (**e**) Time-resolved photoheating of WSe_2_ and Si colloids irradiated by the 830 nm laser; both heating (laser is on) and cooling (laser is off) steps of the experiment are shown. (**f**) Optical extinction and absorption curves of WSe_2_ colloid from (**d**), prepared for the photoheating experiment with a tunable laser source. Absorption curve is calculated using photothermal conversion coefficients, indicated with black arrows and obtained experimentally from the photoheating experiments at different laser wavelengths (see the main text).

**Figure 5 nanomaterials-15-00004-f005:**
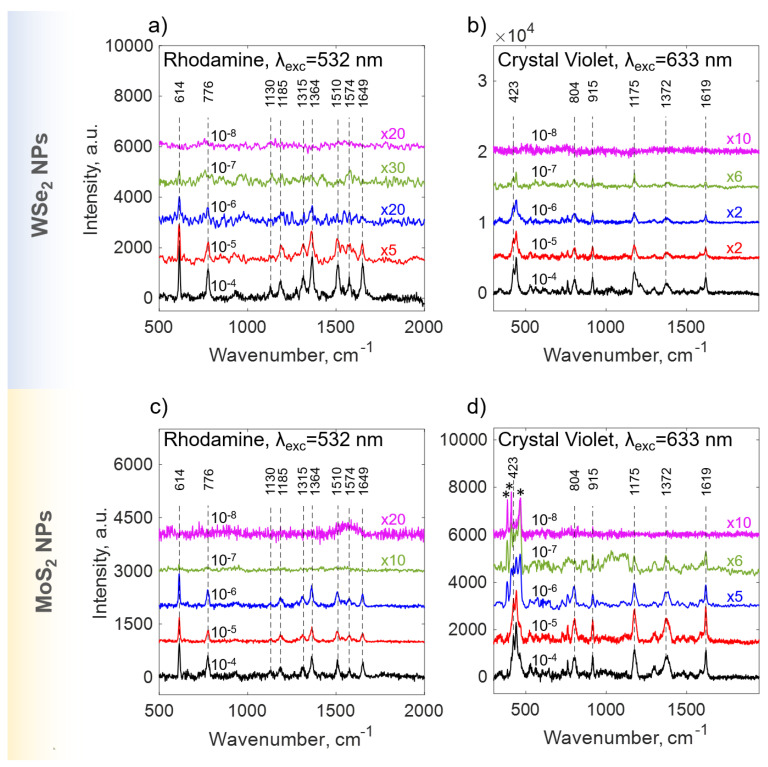
SERS spectra of (**a**) rhodamine 6G (R6G) and (**b**) crystal violet (CV) in the concentration range $10^{-4}$–$10^{-8}$ M adsorbed on WSe_2_ NPs substrate; (**c**) R6G and (**d**) CV in the concentration range $10^{-4}$–$10^{-8}$ M adsorbed on MoS_2_ NPs substrate. Peaks marked by asterisks (*) correspond to MoS_2_.

## Data Availability

The original contributions presented in the study are included in the article and Appendix A; further inquiries can be directed to the corresponding author.

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
