# Peer review of "Tungsten Diselenide Nanoparticles Produced via Femtosecond Ablation for SERS and Theranostics Applications"

_nanomaterials, 2024, doi:10.3390/nano15010004_

Round 1

Reviewer 1 Report

Comments and Suggestions for Authors

Author Response

Comment 1:
In conclusion: the authors have used a fs laser system to fabricate the Tungsten diselenide nanoparticles, and studied their properties and applications. the results are convincing and the discussions are reasonable. some places have English gramar questions, please check to correct.

Response 1:

We thank the Reviewer for the appreciation of our work. We have made additional efforts to improve the readability and make the grammar check of the Manuscript text.

Reviewer 2 Report

Comments and Suggestions for Authors

The manuscript is devoted to laser ablation of 2D TMDC NPs, demonstrating remarkable optical properties for both light absorption and sensing.

The work definitely leaves a positive impression after reading. This opinion is based both on the quality and novelty of the scientific results/conclusions, as well as on the visual presentation.

The Reviewer would like to clarify some points before accepting the manuscript:

1. Based on the data in Fig. 2e,f, it can be said that the obtained NPs are quite polycrystalline. Can we assume in this case that n and k for the NPs are the same as for the bulk (in Fig. 3d)? Accordingly, is this taken into account in the scattering calculations (Fig. 3c)?

2. The experiments in Fig. 4a,b have been performed for the NPs and a bulk on a metal substrate. How does the contact area of ​​the NPs affect the achieved temperature (taking into account that the laser heating was carried out in the equilibrium regime and lasted for a relatively long time)? Is it possible to assert that small contact area of ​​the NPs with the substrate reduces the diffusion of heat into the metal substrate; and thus we observe a dependence as in Fig. 4b? Also, is it possible to somehow separate (estimate) the contributions of "an enhanced absorption cross-section of the NPs" and "the features of a heat diffusion" to the heating efficiency in Fig. 4b? 

3. Additional comments: The meaning of this paragraph is not entirely clear "In such a way, the possibility to tune the bandgap and optical absorption via the appropriate choice of TMDC NPs, as well as their high colloidal stability[30] may be desirable in applications." Also, page 4, lines 118 and 120: "EDX spectrum (Fig. 1c)..." and "SAED pattern (Fig. 1d)..." maybe Figures 2c and 2d?

Author Response

Reviewer's summary:

The manuscript is devoted to laser ablation of 2D TMDC NPs, demonstrating remarkable optical properties for both light absorption and sensing.

The work definitely leaves a positive impression after reading. This opinion is based both on the quality and novelty of the scientific results/conclusions, as well as on the visual presentation.

Our reply:

We thank the Reviewer for the appreciation of our work. Please find the detailed responses below, including the corresponding corrections of the Manuscript.

Comment 1:

Based on the data in Fig. 2e,f, it can be said that the obtained NPs are quite polycrystalline. Can we assume in this case that n and k for the NPs are the same as for the bulk (in Fig. 3d)? Accordingly, is this taken into account in the scattering calculations (Fig. 3c)?

Our reply:

We are grateful to the Reviewer for a constructive question. Indeed, in the Manuscript we demonstrate the polycrystalline structure of the obtained NPs. We take it into account in our scattering simulations (Fig.3c) by choosing a homogenized dielectric function εav=2/3 εo+1/3 εe, where εo and εe are in-plane and out-of-plane dielectric permittivities.

In addition to the information we provided in Fig.3 legend, we add the following phrase into the main text (Page 6, lines 157-160):

The polycrystalline nature of NPs is taken into account in extinction calculations by using a homogenized dielectric function: εav = 2εo / 3 + εe/ 3, where εo and εe are in-plane and out-of-plane bulk dielectric permittivities, correspondingly.

Comment 2:

The experiments in Fig. 4a,b have been performed for the NPs and a bulk on a metal substrate. How does the contact area of ​​the NPs affect the achieved temperature (taking into account that the laser heating was carried out in the equilibrium regime and lasted for a relatively long time)? Is it possible to assert that small contact area of ​​the NPs with the substrate reduces the diffusion of heat into the metal substrate; and thus we observe a dependence as in Fig. 4b? Also, is it possible to somehow separate (estimate) the contributions of "an enhanced absorption cross-section of the NPs" and "the features of a heat diffusion" to the heating efficiency in Fig. 4b?

Our reply:

We thank the Reviewer for the question about the heat transfer to the substrate. In our case we used glass substrate instead of a metallic substrate for precisely this reason to avoid heat transfer to the substrate. Accordingly, we modified the manuscript:

(Page 3, lines 88-90):

For temperature-dependent Raman characterization a drop 20 μl of a sample colloid was drop-casted on a cover glass substrate instead of a metallic substrate to avoid the heat transfer to the substrate and dried at ambient conditions during 1 hour.

(Page 7, lines 175-177):

Figure 4a demonstrates a temperature-dependent shift  in range from 25 °C up to 300 °C of Raman E12g peak of WSe2 NPs, drop-casted on a cover glass substrate instead of a metallic substrate to avoid the heat transfer to the substrate.

Comment 3:

Additional comments: The meaning of this paragraph is not entirely clear "In such a way, the possibility to tune the bandgap and optical absorption via the appropriate choice of TMDC NPs, as well as their high colloidal stability[30] may be desirable in applications." Also, page 4, lines 118 and 120: "EDX spectrum (Fig. 1c)..." and "SAED pattern (Fig. 1d)..." maybe Figures 2c and 2d?

Our reply:

Indeed, the paragraph needs an improvement. Firstly, we logically placed it after the next paragraph, which continues the discussion of SERS results. Secondly, we modified the text (see page 10, lines 252-254 of the improved Manuscript):

In the broader context, PLAL-synthesized colloids from different TMDCs allow to find appropriate optical properties (i.e., refractive index, excitonic absorption), which, considering a high stability of NP solutions [33], may be desirable in applications.

Thank you for indicating the wrong figure numbers. We corrected it and additionally checked the figures referencing across the Manuscript.

Reviewer 3 Report

Comments and Suggestions for Authors

Thanks for the interesting paper. The authors adapted a femtosecond laser ablation for the synthesis of WSe2 nanoparticles (NPs) with diameters from 5 to 150 nm, which conserve the crystalline structure of the original bulk crystal. The research has a good depth of work and is able to unearth many new phenomena. Considering the integrity of the manuscript, the following comments are for reference.

1.      Considering the possibility of the research work being taken and discovered by other readers, it is recommended that the authors add a detailed description of the ultrafast laser processing system. For example, how the nanoparticles are deposited to obtain them and how the laser parameters are set.

2.      For the introduction part, please add some more recent references as a literature survey in the introduction part. Also, the core work of the manuscript relates to the third paragraph of the section, but the author's introduction is too brief, and there is a complete lack of contrast to highlight the necessity and difficulty of the research work. It is recommended that the authors expand on this. For the study of ultrafast laser processing and its applications, the authors may refer to these recent papers: Bioinspired near-full transmittance MgF2 window for infrared detection in extremely complex environments, High Efficiency Femtosecond Laser Ablation of Alumina Ceramics under the Filament Induced Plasma Shock Wave, Damage performance of alumina ceramic by femtosecond laser induced air filamentation, Low-temperature sinterability of graphene-Cu nanoparticles: Molecular dynamics simulations and experimental verification.

3.      The authors said “Previously, we have success- 50 fully used PLAL for preparing NPs of Si [33], TiN [34], MoS2 [30,31] and other materials, 51 including core-satellite nanocomposites [35,36].”. So in that manuscript, the author just switched material? The innovativeness needs to be further highlighted.

4.      The English language and writing standards of the manuscript need to be further improved. For example, the pulsed laser ablation in liquids in line 62.

5.      For the EDX results in the Figs. 1 and 2, it is recommended that the authors add an analysis of elemental content.

6.      I have one of the biggest doubts. How were the laser parameters used determined in the author's manuscript. It is recommended that the authors discuss and analyze this. How the laser parameters affect the material properties and how to go about modulating the material properties. This will show the significance of the manuscript. For example, is it possible to further optimize the size distribution uniformity of the NPs. Is it possible to obtain the corresponding performance or even higher with other laser parameters?

Comments on the Quality of English Language

The English could be improved to more clearly express the research.

Author Response

Reviewer's summary:

Thanks for the interesting paper. The authors adapted a femtosecond laser ablation for the synthesis of WSe2 nanoparticles (NPs) with diameters from 5 to 150 nm, which conserve the crystalline structure of the original bulk crystal. The research has a good depth of work and is able to unearth many new phenomena.

Our reply:

We thank the Reviewer for an appreciation of our work. Please find below the Reviewer's comments and our replies, including necessary corrections in the Manuscript and Supplementary Materials.

Comment 1:

Considering the possibility of the research work being taken and discovered by other readers, it is recommended that the authors add a detailed description of the ultrafast laser processing system. For example, how the nanoparticles are deposited to obtain them and how the laser parameters are set.

Our reply:

We thank the Reviewer for a thoughtful comment. We added the information about the laser parameters (such as wavelength, pulse energy, repetition rate) and experimental setup into the Section 2 of the Manuscript (subsection 2.1 “Sample preparation”). In addition, we added a detailed scheme of the optical setup in Supplementary Materials, Note 1. In order to clarify, how NPs were deposited for the temperature-dependent Raman characterization, we added a paragraph on Page 3, lines 88-90:

For temperature-dependent Raman characterization a drop 20 μl of a sample colloid was drop-casted on a cover glass substrate instead of a metallic substrate to avoid the heat transfer to the substrate and dried at ambient conditions during 1 hour.

Comment 2:

For the introduction part, please add some more recent references as a literature survey in the introduction part. Also, the core work of the manuscript relates to the third paragraph of the section, but the author's introduction is too brief, and there is a complete lack of contrast to highlight the necessity and difficulty of the research work. It is recommended that the authors expand on this. For the study of ultrafast laser processing and its applications, the authors may refer to these recent papers: Bioinspired near-full transmittance MgF2 window for infrared detection in extremely complex environments, High Efficiency Femtosecond Laser Ablation of Alumina Ceramics under the Filament Induced Plasma Shock Wave, Damage performance of alumina ceramic by femtosecond laser induced air filamentation, Low-temperature sinterability of graphene-Cu nanoparticles: Molecular dynamics simulations and experimental verification.

Our reply:

We thank the Reviewer for the suggestion. We improved the Introduction by adding the following paragraph (Page 2, lines 39-41):

Femtosecond laser material processing is a high-throughput manufacturing tool employed in science and industry and offers advantages of a flexible, precise treatment of a wide range of materials [29–31]. 

Comment 3:

The authors said “Previously, we have success- 50 fully used PLAL for preparing NPs of Si [33], TiN [34], MoS2 [30,31] and other materials, 51 including core-satellite nanocomposites [35,36].”. So in that manuscript, the author just switched material? The innovativeness needs to be further highlighted.

Our reply:

We thank the reviewer for the question. In fact, PLAL approach for the synthesis of TMDC NPs is an emerging research field with many unexplored aspects. In more detail we refer to this in our response to the Comment 6 of the Reviewer 3. In addition, various TMDC materials have different ablation thresholds and thermal degradation properties that affect the crystalline structure of resulting NPs. Thus, the PLAL protocol requires a careful adaptation to the certain material.

In addition to the novel material adapted for the PLAL-synthesized NPs, we demonstrate optical constants of WSe2 crystal (the bulk response, not from a single- or several-layered flakes) for the first time, as well as WSe2 colloids photothermal response.

Comment 4:

The English language and writing standards of the manuscript need to be further improved. For example, the pulsed laser ablation in liquids in line 62.

Our reply:

We thank the reviewer for the suggestion. We have made additional efforts to improve the readability and make the grammar check of the Manuscript text.

Comment 5:

For the EDX results in the Figs. 1 and 2, it is recommended that the authors add an analysis of elemental content.

Our reply:

We thank the reviewer for the valuable suggestion. Indeed, the elemental content analysis is required for the EDX results. We made the following additions in the main text (Page 4, line 107):

The Energy-dispersive X-ray spectroscopy (EDX) of the crystal (Fig.1b) confirms the stoichiometry quality: W (32.52 at.%), Se (67.48 at.%)

and (Page 5, lines 126-127):

EDX spectrum (Fig. 2c) confirms that NPs conserve the atomic composition of the original crystal: W (12.95 at.%), Se (29.88 at.%); some additional peaks correspond to the TEM grid copper Cu (57.17 at.%).

Comment 6:

I have one of the biggest doubts. How were the laser parameters used determined in the author's manuscript. It is recommended that the authors discuss and analyze this. How the laser parameters affect the material properties and how to go about modulating the material properties. This will show the significance of the manuscript. For example, is it possible to further optimize the size distribution uniformity of the NPs. Is it possible to obtain the corresponding performance or even higher with other laser parameters?

Our reply:

We thank the Reviewer for the thoughtful comment. The method of pulsed laser ablation in liquids (PLAL) we use in the work allows various pathways for the performance optimization. For example, the choice of specific solvent (deionized water in our case) defines the plasma plume condensation dynamics due to the solvent density, vapor pressure and its cooling rate. The total solvent volume also affects the laser focusing conditions, filament behavior and effective particle concentration during the ablation process, which controls the total light-matter interaction time. In addition, the lens-to-target distance defines the energy flux in the ablation spot.

All the mentioned degrees of freedom affect the important synthesis characteristics: NPs size distribution, crystallinity and shape.

Thus, the optimization of the PLAL synthesis protocol is a challenging task and deserves a separate study. In our experience, a higher numerical aperture lens maximizes the laser fluence and improves the throughput, but simultaneously yields a broader size distribution due to the high fluence gradient. Therefore, to prepare more narrow size distributions, a differential centrifugation method was employed.

The better PLAL performance can be considered in various aspects. For example, the increase of the laser power will definitely improve the ablation rate due to the larger material volume involved in the ablation process. Alternatively, the lens-to-target distance variation affects the NPs crystallinity (amorphous/crystalline structure).

Round 2

Reviewer 3 Report

Comments and Suggestions for Authors

The authors have revised the manuscript accordingly.